# Crosshair, semi-automated targeting for electron microscopy with a motorised ultramicrotome

Kimberly Meechan[1,2]*, Wei Guan[3], Alfons Riedinger[4], Vera Stankova[4], Azumi Yoshimura[3], Rosa Pipitone[1], Arthur Milberger[5], Helmuth Schaar[5], Inés Romero-Brey[1], Rachel Templin[1†], Christopher J Peddie[3], Nicole L Schieber[1‡], Martin L Jones[3], Lucy Collinson[3], Yannick Schwab[1,6]*

[1]Cell Biology and Biophysics Unit, European Molecular Biology Laboratory (EMBL), Heidelberg, Germany; [2]Collaboration for Joint PhD Degree Between EMBL and Heidelberg University, Faculty of Biosciences, Heidelberg, Germany; [3]Francis Crick Institute, London, United Kingdom; [4]Electronic Workshop, European Molecular Biology Laboratory (EMBL), Heidelberg, Germany; [5]Mechanical Workshop, European Molecular Biology Laboratory (EMBL), Heidelberg, Germany; [6]Electron Microscopy Core Facility, European Molecular Biology Laboratory (EMBL), Heidelberg, Germany

*For correspondence:
kimberly.meechan@embl.de (KM);
yannick.schwab@embl.de (YS)

Present address: †Ramaciotti Centre for Cryo-EM, Monash University, Melbourne, Australia; ‡Centre for Microscopy and Microanalysis, University of Queensland, Brisbane, Australia

Competing interest: The authors declare that no competing interests exist.

**Abstract** Volume electron microscopy (EM) is a time-consuming process – often requiring weeks or months of continuous acquisition for large samples. In order to compare the ultrastructure of a number of individuals or conditions, acquisition times must therefore be reduced. For resin-embedded samples, one solution is to selectively target smaller regions of interest by trimming with an ultramicrotome. This is a difficult and labour-intensive process, requiring manual positioning of the diamond knife and sample, and much time and training to master. Here, we have developed a semi-automated workflow for targeting with a modified ultramicrotome. We adapted two recent commercial systems to add motors for each rotational axis (and also each translational axis for one system), allowing precise and automated movement. We also developed a user-friendly software to convert X-ray images of resin-embedded samples into angles and cutting depths for the ultramicrotome. This is provided as an open-source Fiji plugin called Crosshair. This workflow is demonstrated by targeting regions of interest in a series of *Platynereis dumerilii* samples.

## Editor's evaluation

Meechan et al. present a systematic approach to semi-automate an ultramicrotome operation for targeting a specific plane aided by X-ray tomography measurements, and it is a fundamental work of great interest to any users of electron microscopy (EM), particularly when targeting the imaging of thin sections in a select region of interest by ultramicrotomy, or when targeting volume EM of select sample regions. The article documents with exceptional detail a workflow including both microtome modifications and software adaptations for semi-automated targeting of structures with micrometre precision, resulting in a faster and more accurate orientation of the image acquisition planes for volume electron microscopy, a task that has traditionally been difficult and time-consuming. Therefore, this work will reduce sample preparation labour and, critically, facilitate the comparison of the ultrastructure of multiple samples. The method is based on X-ray imaging acquisition prior to any sectioning and proposes a solution for the two instruments commercially available in the field, and by transparently sharing all the data, hardware, and software, and describing every detail of the workflow, this fundamental method can be readily adopted by any practitioner, enabling its wide

application – it is a key step in the field regarding speed-up, accuracy, and reproducibility in electron microscopy.

## Introduction

Imaging samples with electron microscopy (EM) is a time-consuming process – often requiring weeks or months of continuous acquisition for large samples (*Scheffer et al., 2020*; *Zheng et al., 2018*; *Vergara et al., 2021*). This means that it is rarely feasible to acquire image volumes from entire specimens, and we must instead target specific regions of interest. For resin-embedded samples, this process is usually done with an ultramicrotome that allows manual trimming of the block using a razor blade and/or diamond knife, followed by cutting of thin sections at precise locations.

Targeting with an ultramicrotome is a difficult and manual process. The operator must rely on surface features that are visible through the ultramicrotome's binocular microscope. This makes it challenging to target regions deep within a sample, especially when heavy metals make the sample opaque. In addition, the orientation of cutting must be set manually by adjusting three different ultramicrotome axes. This kind of 3D thinking is challenging, taking much time and training to master.

Other studies have improved the precision and ease of targeting by leveraging external 3D maps from light or X-ray microscopy (*Karreman et al., 2016*; *Karreman et al., 2017*; *Xu et al., 2021*; *Musser et al., 2021*; *Bushong et al., 2015*; *Ronchi et al., 2021*). X-ray offers many advantages for EM targeting – for example, it is readily compatible with standard EM sample preparation methods (*Bosch et al., 2022*; *Kuan et al., 2020*; *Bushong et al., 2015*; *Karreman et al., 2017*) and provides images that highlight similar structures to volume EM (although at a lower resolution) (*Bosch et al., 2022*; *Kuan et al., 2020*; *Bushong et al., 2015*; *Musser et al., 2021*). Laboratory micro-CT systems are becoming ever more popular and can provide an isotropic resolution of about 1 micron, with scan times in the range of a few hours (*Withers et al., 2021*; *Karreman et al., 2016*). In addition, there is increasing access to synchrotrons for X-ray imaging, which can provide resolutions in the range of hundreds or even tens of nanometres, and scan times in the range of minutes (depending on the resolution required) (*Withers et al., 2021*; *Bosch et al., 2022*; *Kuan et al., 2020*). X-ray imaging therefore offers a fast, non-destructive method for obtaining 3D maps of the internal features of a sample for targeting.

Most methods that use light or X-ray microscopy for targeting rely on measuring the depth of regions of interest from the resin block's surface, followed by trimming with an ultramicrotome. As these methods do not compensate for the exact position and orientation of the sample in the ultramicrotome, progress must be checked at regular intervals by iterative rounds of X-ray or light microscopy and trimming, slowing this process down. In addition, these methods focus on the depth of the region of interest, but still require the orientation of cutting to be set entirely manually.

Other studies have sought to bring more automation to the ultramicrotome to make it easier to use (*Baena et al., 2019*; *Templier, 2019*; *Lee et al., 2018*). These efforts have mostly focused on the collection of serial sections, with little automation of the initial trimming and targeting process. An exception is *Brama et al., 2016*, in which a miniature fluorescence microscope was integrated into the ultramicrotome, allowing automated tracking of regions of interest during cutting. However, this requires a fluorescent signal to be present within the sample embedded in the resin block and is therefore not compatible with standard heavy-metal stained specimens. Also, while this technique allows detection and collection of sections from regions of interest, it does not automate the depth or orientation of ultramicrotome cutting.

Here, we created a workflow for semi-automated targeting of regions of interest with a modified ultramicrotome. We introduced automation to two recent commercial systems in the form of motorisation for each of the ultramicrotome axes. This allowed precise and automated angular movement for both systems, as well as translational movement for one system. We also created new software that allows selection of a plane of interest from X-ray images, and automatic conversion into angles and cutting depths for the ultramicrotome. This is provided as an open-source, user-friendly Fiji (*Schindelin et al., 2012*) plugin called Crosshair.

# Results

## Automation of ultramicrotome

We modified two of the most recent commercial ultramicrotomes (Leica EM UC7 and RMC Power-Tome PC [PTPC]) to add automation of movement (*Figure 1*, *Figure 1—figure supplement 1*). For the UC7, this consisted of three extra motors (one for each rotational axis) controlled by a small Raspberry Pi computer (*Figure 1B and D*). This system can be controlled via a simple touchscreen interface (developed with Node-RED), allowing the angles of each axis to be precisely set and measured (see section 'Ultramicrotome systems and calibration'). All translational movements of the knife (north–south [NS] and east–west [EW]) were controlled manually via the existing Leica stage motors and Leica user interface.

For the RMC, motors were added to the same rotational axes, with additional automation of the two translational stage movements (*Figure 1C and E*). These additional NS and EW motors allowed the position of the knife to be precisely controlled within the same user interface as the orientation. This is in contrast to the Leica system, where currently the translational and rotational movements are controlled separately. The five motors on the RMC system were controlled by a Trinamic TMCM-6110 6-axis stepper motor driver board. The graphical user interface (GUI) was developed with LabVIEW and installed onto a touchscreen PC, allowing users to control the motions via keyboard/mouse, touchscreen, or a joystick (see section 'Ultramicrotome systems and calibration').

Both systems were calibrated with special adaptors made to convert a rotational movement into the rotation of the dial of a Thorlabs CRM1PT/M rotation mount, whose Vernier scale provides 5 arcmin (0.083°) resolution (see section 'Ultramicrotome systems and calibration).

## Targeting calculations

Our goal was to allow automatic calculation of the ultramicrotome moves required to cut to a specific target plane. The target plane defined the final position and orientation of the desired block surface (*Figure 2A*). Thin sections could then be taken for transmission electron microscopy (TEM) or array tomography or the surface could be imaged directly with volume scanning electron microscopy (SEM).

This targeting problem was broken down into two steps: orientation and distance. For the orientation problem, we found a combination of three angles (knife tilt, sample tilt, sample rotation; *Figure 1A*) that brought the target plane to be vertical and parallel to the knife (*Figure 2B*). For the distance problem, we found the depth to cut from the sample surface to reach the target plane (*Figure 2C*).

The orientation problem required a mathematical description of how the target plane orientation changed as the knife tilt, sample tilt, and sample rotation were varied. This was a similar problem to that which must be solved to, for example, orient the 'end-effector' of a robotic arm by varying the angle of its individual joints. In robotics, this is solved by construction of a 'forward kinematics' equation that calculates the end-effector position, given the angles of the joints, and an 'inverse kinematics' equation, which calculates the joint angles, given the desired end-effector position. We therefore applied similar principles, constructing a forward and inverse kinematics equation for the ultramicrotome.

We first constructed the forward kinematics equation to describe the orientation of the target plane in terms of the three input angles. To do so, we defined orthogonal coordinate frames for each of the ultramicrotome parts (e.g. the knife holder, the sample holder, etc.) (*Figure 3A and B*, *Figure 3—figure supplement 1*) and calculated the rotations that relate each piece. This is summarised in the pose diagram in *Figure 3C*. The relation between the sample holder and sample block surface is unknown in this pose diagram. This is because the sample holder does not hold the sample in a fixed orientation, meaning that the resin block has a slightly different orientation every time it is placed inside.

To calculate this relation, we therefore required an initial alignment step where the knife is manually aligned to the block face. This is standard procedure for cutting with an ultramicrotome, and so should be familiar to anyone who has used a ultramicrotome previously. In this aligned position, the orientation of the knife and block coordinate frames was the same, allowing the holder to block relation to be calculated (*Figure 3D*).

This meant that the full forward kinematics equation, describing the rotation from the World coordinate frame to the target plane, was

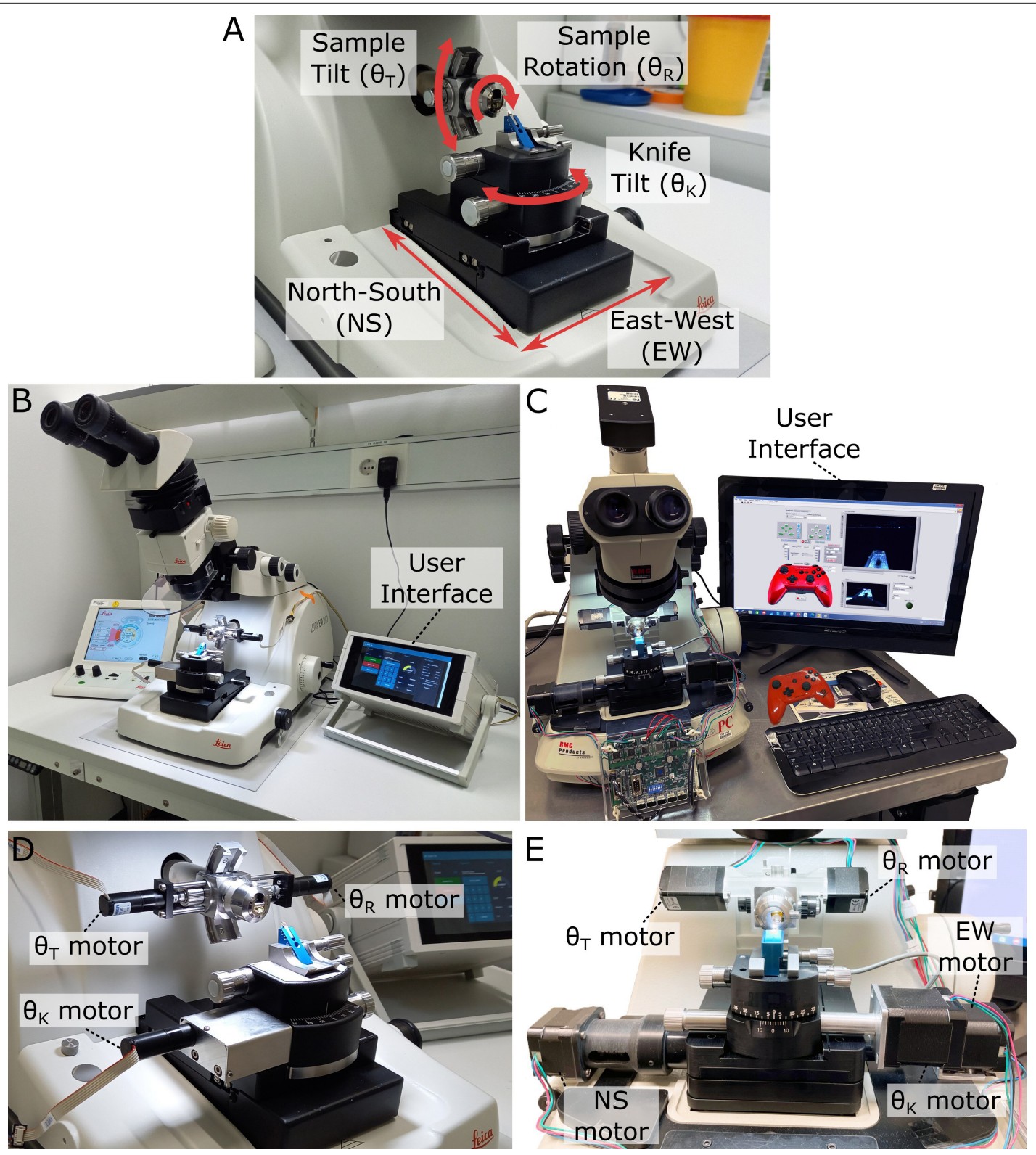

**Figure 1.** Motorised ultramicrotome. (**A**) Summary of the main ultramicrotome axes of rotation (sample tilt, sample rotation, and knife tilt), and movement (north–south [NS]/east–west [EW]), which are common to both Leica and RMC systems. (**B**) Overview of motorised Leica system. (**C**) Overview of motorised RMC system. (**D**) Zoom of (**B**), showing motors attached to each axis. (**E**) Zoom of (**C**), showing motors attached to each axis.

The online version of this article includes the following figure supplement(s) for figure 1:

*Figure 1 continued on next page*

**Figure supplement 1.** Motorised ultramicrotomes and user interfaces.

**Figure supplement 2.** Calibration of sample tilt axis.

**Figure supplement 3.** Calibration of sample rotation axis.

**Figure supplement 4.** Calibration of knife tilt axis.

$$F = R_x(\theta_T)R_y(\theta_R)R_x(-\theta_{IT})R_z(\theta_{IK})R_z(\theta_{to})R_x(\theta_{tr})$$

where $R_x$, $R_y$, and $R_z$ are rotation matrices about the x, y, and z axes, respectively. $\theta_T$ is the sample tilt angle and $\theta_R$ is the sample rotation angle. $\theta_{IT}$, $\theta_{IK}$, $\theta_{to}$, and $\theta_{tr}$ are constants for a particular targeting run representing the initial sample tilt angle (on alignment), the initial knife tilt angle (on alignment), the offset angle between the block face and the target plane (measured from X-ray), and the rotation angle between the block face and the target plane (measured from X-ray). See *Figure 3B* for details of how $\theta_{to}$ and $\theta_{tr}$ are calculated.

With the forward kinematics equation complete, we then constructed and solved an inverse kinematics equation to determine the required angles for each solution (see section 'Orientation calculation'). This gave the solution tilt and knife angles as

$$\theta_T = \arctan\left(C_1\cos(\theta_R) + C_2\sin(\theta_R)\right)$$

$$\theta_K = \arctan\left(\frac{C_3\left(C_4\cos(\theta_R) + C_5\sin(\theta_R)\right)}{\left(\sqrt[2]{C_3{}^2 + \left(C_4\sin(\theta_R) - C_5\cos(\theta_R)\right)^2}\right)|C_3|}\right)$$

where $C_{1-5}$ are constants for a particular targeting run (see 'Materials and methods' for details). This means that, as expected, there are many possible solutions for each targeting setup. In fact, any point on these lines (*Figure 3E*) is a possible solution. Note that there will be some situations where no solution is possible due to the constraints of the ultramicrotome axes, for example, in *Figure 3F*, the sample tilt of some solutions exceeds the maximum sample tilt of 20°. In these situations, the sample would need to be remounted or re-trimmed to aim for an angular difference within the ultramicrotome constraints.

For the distance problem, we calculated the distance between the block surface and target plane (from the X-ray images), and compensated for the current knife angle to give the true cutting distance:

$$D_{NS} = \frac{D_P}{\cos(\theta_K)}$$

where $D_{NS}$ is the north–south (NS) distance (cutting distance), $D_P$ is the perpendicular distance between the target plane and the furthest surface point, and $\theta_K$ is the knife angle (see section 'Distance calculation' and *Figure 3—figure supplement 2*).

## Crosshair targeting workflow

The entire workflow is summarised in *Figure 4*, with details given in the section 'Targeting workflow'. In brief, the resin block was first trimmed manually close to the sample, with a rectangular face in its surface. This block face (which can be any shape with four corners and straight lines in between) provided the reference to define the block coordinate frame (*Figure 3A*).

Next, the sample was imaged with X-ray at the highest resolution possible (around 1 micron isotropic voxel size is feasible on lab-based micro-CT systems). From this X-ray image, two planes were defined: one on the flat block surface, and one on the target plane using the Crosshair plugin (see section 'Crosshair' and *Figure 2A*). Then, the corners of the block surface were marked in the software as these were used to determine where cutting would begin and the distance required. Finally, the orientation was decided, that is, which edge of the block would face upwards in the ultramicrotome.

Next, we moved to the ultramicrotome. The zero position of the sample tilt and knife tilt axes was checked (see 'Materials and methods') to ensure all angular measures began from the correct

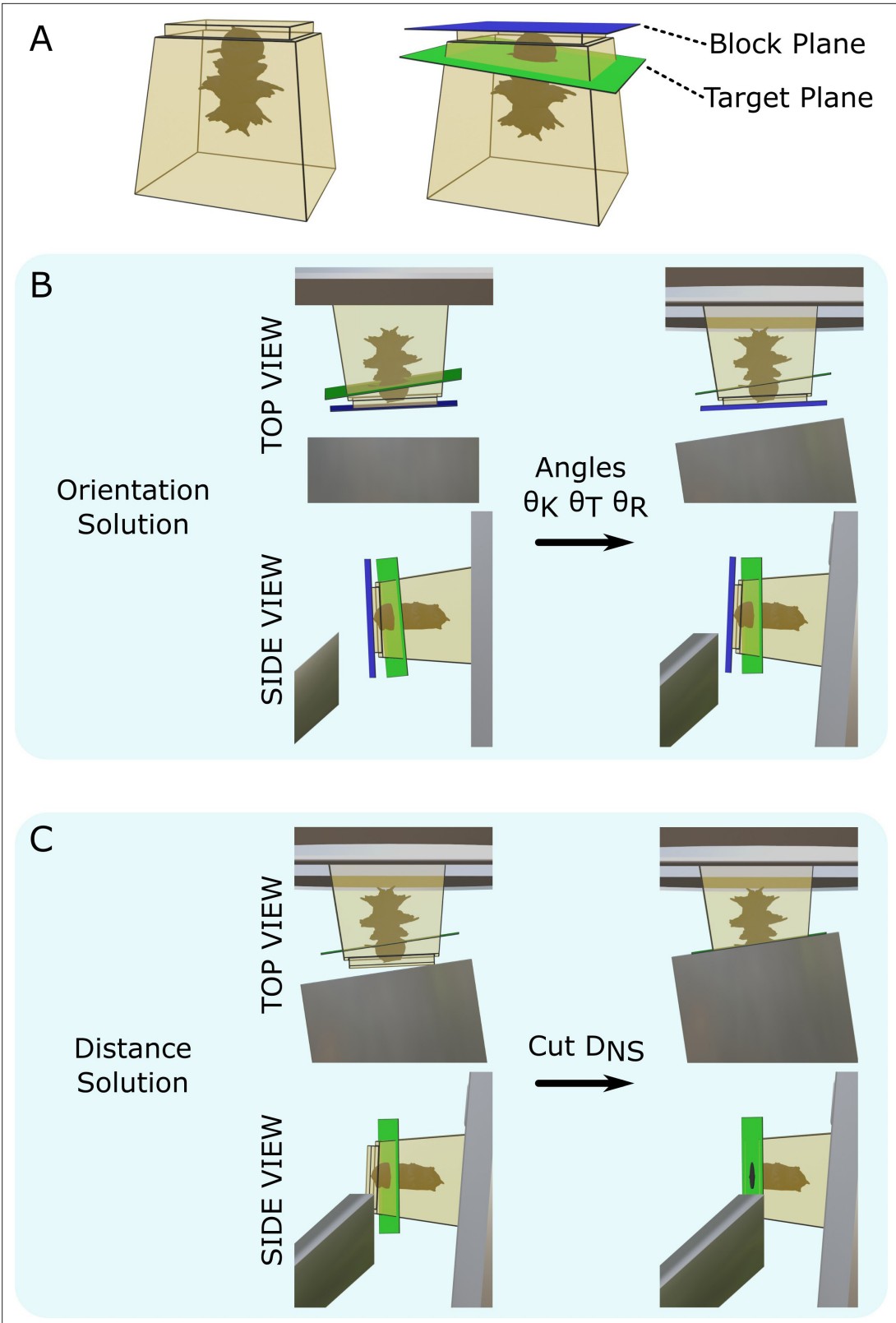

**Figure 2.** Targeting problem. (**A**) Left: diagram of a resin block with a *Platynereis* sample inside. The top surface has been trimmed flat with a diamond knife. Right: same resin block with labelled block plane (i.e. the plane parallel to the flat trimmed surface) and target plane. (**B**) Diagram of the block in the ultramicrotome before (left) and after (right) the orientation solution is applied. A top view (with diamond knife at the bottom, and sample holder at the top) and side view (with diamond knife on the left, and sample holder on the right) are shown. $\theta_K$ is the knife tilt angle, $\theta_T$ is the sample tilt angle,

*Figure 2 continued on next page*

*Figure 2 continued*

and $\theta_R$ is the sample rotation angle. (**C**) Diagram of the block in the ultramicrotome before (left) and after (right) the distance solution is applied. Top/side view are the same as in (**B**). $D_{NS}$ is the NS cutting distance.

position. Then, the resin-embedded sample was inserted, and the knife and block face manually aligned. This aligned position was defined as the zero position for the sample rotation to simplify the targeting calculations.

The aligned sample and knife tilt angles ($\theta_{IT}$ and $\theta_{IK}$) were then precisely read from the motorised ultramicrotome system and entered into the targeting calculations. This allowed all solutions to be calculated and one selected by the ultramicrotome user. For example, the user may want to minimise the knife and sample tilt angles to make it easier to collect thin sections after targeting. Finally, the ultramicrotome was precisely set to the three solution angles using the motors and the solution distance cut.

At this point, thin sections could be taken for TEM, or array tomography, or the block could be transferred for volume imaging in a focused ion beam scanning electron microscope (FIB-SEM) or serial block face scanning electron microscope (SBF-SEM).

## Crosshair

To make this workflow accessible to all, we developed the open-source Fiji (*Schindelin et al., 2012*) plugin Crosshair (https://github.com/automated-ultramicrotomy/crosshair; *Meechan, 2022b*), which allows user-friendly measurement of X-ray images, and access to the calculations described above (*Figure 5*, *Figure 5—figure supplement 1*). It uses BigDataViewer (*Pietzsch et al., 2015*) to allow browsing of large images in arbitrary 2D slices, and the ImageJ 3D Viewer (*Schmid et al., 2010*) to allow browsing in 3D.

Crosshair allows the planes and points required for the targeting workflow (*Figure 4*) to be created and viewed interactively. It also allows the orientation of the block to be marked and visualised in 2D and 3D.

To make selection of solutions easier, Crosshair also has a 'microtome mode' which allows a simple representation of the ultramicrotome to be manipulated in 3D (*Figure 5B*). This has the constraints of the ultramicrotome built in and displays a volume rendering of the X-ray imaged sample at the appropriate orientation. In this way, users can interactively move sliders in the user interface and view the orientation of the ultramicrotome and sample in real time. This also outputs the corresponding angles and cutting distance for each solution.

Finally, Crosshair also has a 'cutting mode' which allows the predicted cutting progression to be viewed in 2D and 3D (*Figure 5—figure supplement 1C*). In this way, users can choose a solution and then easily view which structures will be visible at which cutting distances. This makes it simple to choose an appropriate solution for their targeting problem.

## Accuracy measures

To test the accuracy of this workflow, we targeted a series of 6-day-old *Platynereis dumerilii* samples. *Platynereis* is a marine annelid, which is an established model system for development, evolution, and neurobiology (*Özpolat et al., 2021*; *Fischer and Dorresteijn, 2004*; *Williams and Jékely, 2016*). As they can be bred quickly and easily in the lab, and have well-defined tissue morphology, we decided to use them as a test case for targeting regions of interest. We set our target plane on each sample to cut through both of the anterior dorsal cirrus that protrude from either side of the *Platynereis* head.

All samples were targeted with the workflow described above: five on the motorised Leica system (samples a–e) and five on the RMC system (samples f–j). After targeting, samples were X-ray imaged again and registered to the pre-targeting scan. This registration was completed with another Fiji plugin we created – RegistrationTree – which is also freely available on GitHub (https://github.com/K-Meech/RegistrationTree; *Meechan, 2022c*). This plugin is a wrapper around two existing registration software – BigWarp (*Bogovic et al., 2016*) and elastix (*Klein et al., 2010*; *Shamonin et al., 2013*) – allowing easy passing of images from one software to the other, and smooth use of very large images with elastix. It is standalone, so it can also be used for general registration of large images, independent of the Crosshair workflow.

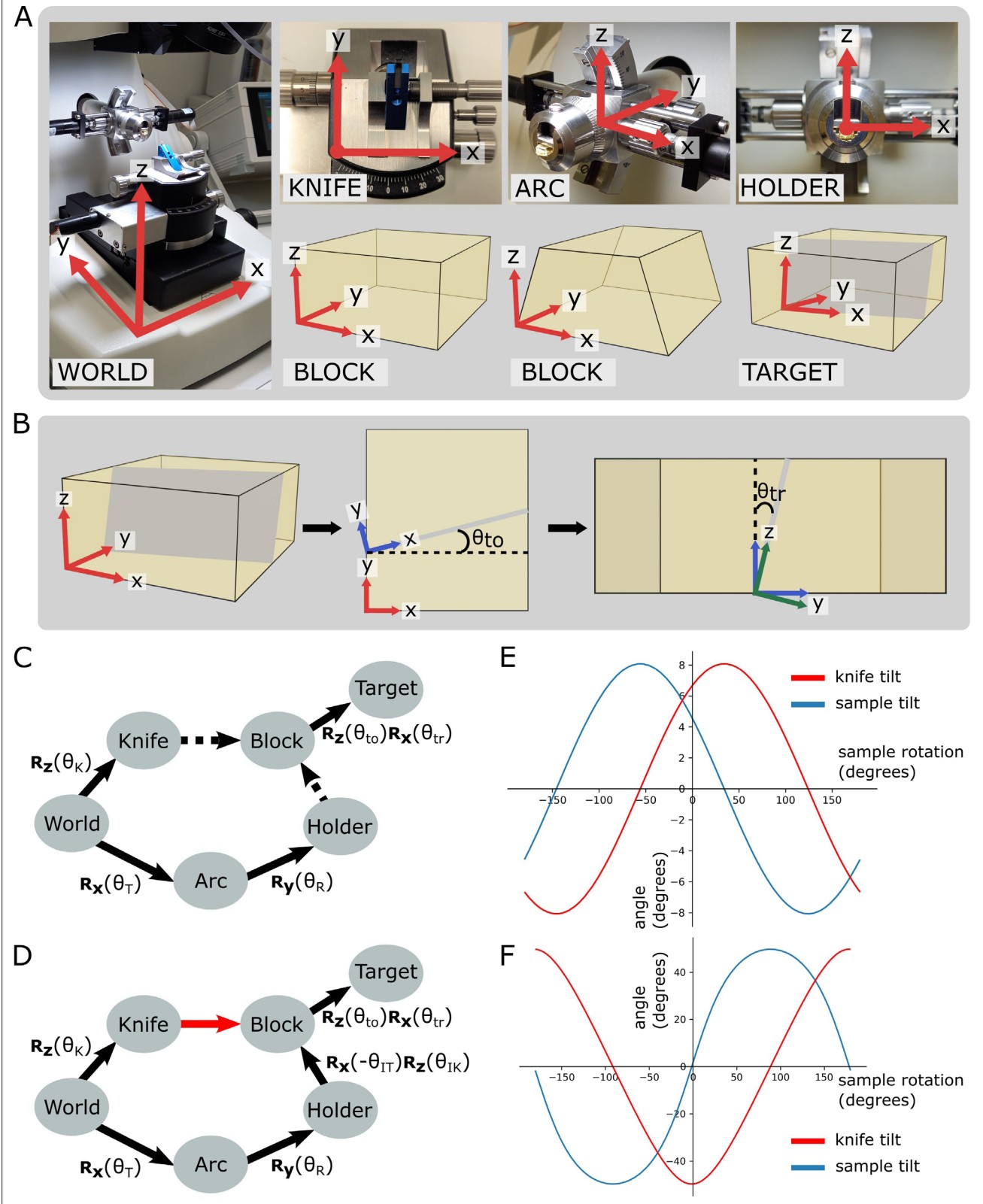

**Figure 3.** Coordinate frames, pose diagrams, and solutions. (**A**) Diagram of all coordinate frames. World: x is parallel to EW, y is parallel to NS, z is vertical. Knife: z points out of the page, and is parallel to the world frame z. Arc: x is parallel to the world frame x. Holder: y points into the page and is parallel to the arc frame y. Block: x is parallel to the bottom edge of the block face. Two variations are shown: a rectangular block face (left) and a trapezium block face (right). Target: y is perpendicular to the target plane (shown in grey). (**B**) Diagram of relation between block and target coordinate

*Figure 3 continued on next page*

*Figure 3 continued*

frames. Left: 3D view of the block, showing the block frame (red) and target plane (grey). Middle: 2D view of the block frame's xy plane (i.e. looking down on the block from above). The target plane intersects along the shown grey line, and $\theta_{to}$ is the angle between the red and blue x axes, that is, the rotation angle about the block frame z. Right: 2D view of the blue axes' zy plane (i.e. looking down the line of intersection from the middle panel). $\theta_{tr}$ is the angle between the blue and green z axes, that is, the rotation angle about the blue frame x. The green axes are the final target coordinate frame (as shown in **A**). (**C**) Pose diagram showing relations between coordinate frames. $\boldsymbol{R_x}$, $\boldsymbol{R_y}$, $\boldsymbol{R_z}$: rotation matrices about the x, y, and z axes, respectively; $\theta_T$: sample tilt angle; $\theta_R$: sample rotation angle; $\theta_K$: knife angle; $\theta_{to}$: target offset angle; $\theta_{tr}$: target rotation angle. Dashed arrows are unknown relations. (**D**) Pose diagram when the knife is aligned to the block face. Here, the knife frame is in the same orientation as the block (red arrow), and therefore we can infer the Holder to Block relation that was unknown in (**C**). $\theta_{IT}$ is the initial sample tilt angle and $\theta_{IK}$ is the initial knife tilt angle, when the knife and block are aligned. We define $\theta_R$ to be zero at this aligned orientation. (**E**) Graphs of knife and sample tilt solution for all sample rotation angles. This used values of $\theta_{IK} = 10$, $\theta_{IT} = 10$, $\theta_{to} = -3.3$, and $\theta_{tr} = 5.4$. (**F**) Solution graphs with more extreme initial angles. $\theta_{IK} = 10$, $\theta_{IT} = 10$, $\theta_{to} = -60$, and $\theta_{tr} = 5.4$.

The online version of this article includes the following figure supplement(s) for figure 3:

**Figure supplement 1.** Coordinate frames at different angles.

**Figure supplement 2.** Distance calculation.

Once the images were registered, we calculated a series of angle and distance accuracy measures using the built-in features in the Crosshair plugin (*Figures 6 and 7*; see section 'Accuracy tests').

Comparison of the target plane and trimmed block surface showed that all samples hit their target of cutting through both anterior dorsal cirrus, and provided similar *Platynereis* cross-sections to those predicted from the X-ray (*Figure 7*). The accuracy measures gave a mean angular error of 0.9°, a mean absolute solution distance error of 3.1 microns, and a mean point-to-plane distance error of 3.1 microns (see section 'Accuracy tests' for details of these different measures). Note that there will be some error in these distance measures due to slight inaccuracies in the registration process – estimated at around 2 microns error (see section 'Accuracy tests' and *Figure 6—figure supplement 1*). Given that our X-ray voxel size is only 1 micron, there is a limit to how precisely such small distances can be measured. This could be improved in future by moving to higher-resolution X-ray imaging from synchrotrons, which can provide resolutions in the range of hundreds of nanometres.

Note that there were also some differences between the mean targeting accuracy reached with the Leica and RMC systems. The Leica system had a mean angular error of 1.1°, mean absolute solution distance error of 4.5 microns, and a mean point-to-plane distance error of 4.8 microns. The RMC system had a mean angular error of 0.7°, mean absolute solution distance error of 1.6 microns, and a mean point-to-plane distance error of 1.3 microns. There are a number of factors that could have contributed to this difference. Firstly, the samples tested with the Leica system were X-ray imaged with a different micro-CT system to those tested on the RMC system (see section 'X-ray imaging'). While all were scanned with the same voxel size (1 micron), the amount of detail that could be seen inside the *Platynereis* was quite different (*Figure 7—figure supplement 1*). Therefore, the higher quality of the X-ray scans for samples f–j (tested with RMC) may have contributed to the improved average targeting accuracy. Secondly, there are still a number of manual steps in this workflow (such as the manual alignment of knife and block face), which can allow human error to contribute to the accuracy. Samples a–e and f–j were targeted by different researchers, so this may also contribute to the differing average accuracy. Finally, the Leica and RMC systems use different hardware (e.g. motors) which will also contribute to their differing average accuracy.

## Discussion

The motorised ultramicrotome systems and targeting workflow presented here provide a user-friendly solution for targeted EM. The workflow allows X-ray images of resin-embedded samples to be converted into a set of three angles (sample tilt, sample rotation, and knife tilt) and a cutting distance for the ultramicrotome. These moves can then be precisely executed with the two motorised ultramicrotome systems we developed to reach a target plane. All measurement steps are integrated into the open-source Fiji plugin Crosshair, making it easily accessible to all. Across 10 targeting runs, the workflow provided a mean angular error of 0.9° and a mean distance error of 3.1 microns. Once the X-ray images have been acquired, the Crosshair measurements and trimming can be done in 1–2 hr, depending on the sample. This is a significant speed up where, in our experience, non-assisted targeting from X-ray may take 1–2 days for complex samples.

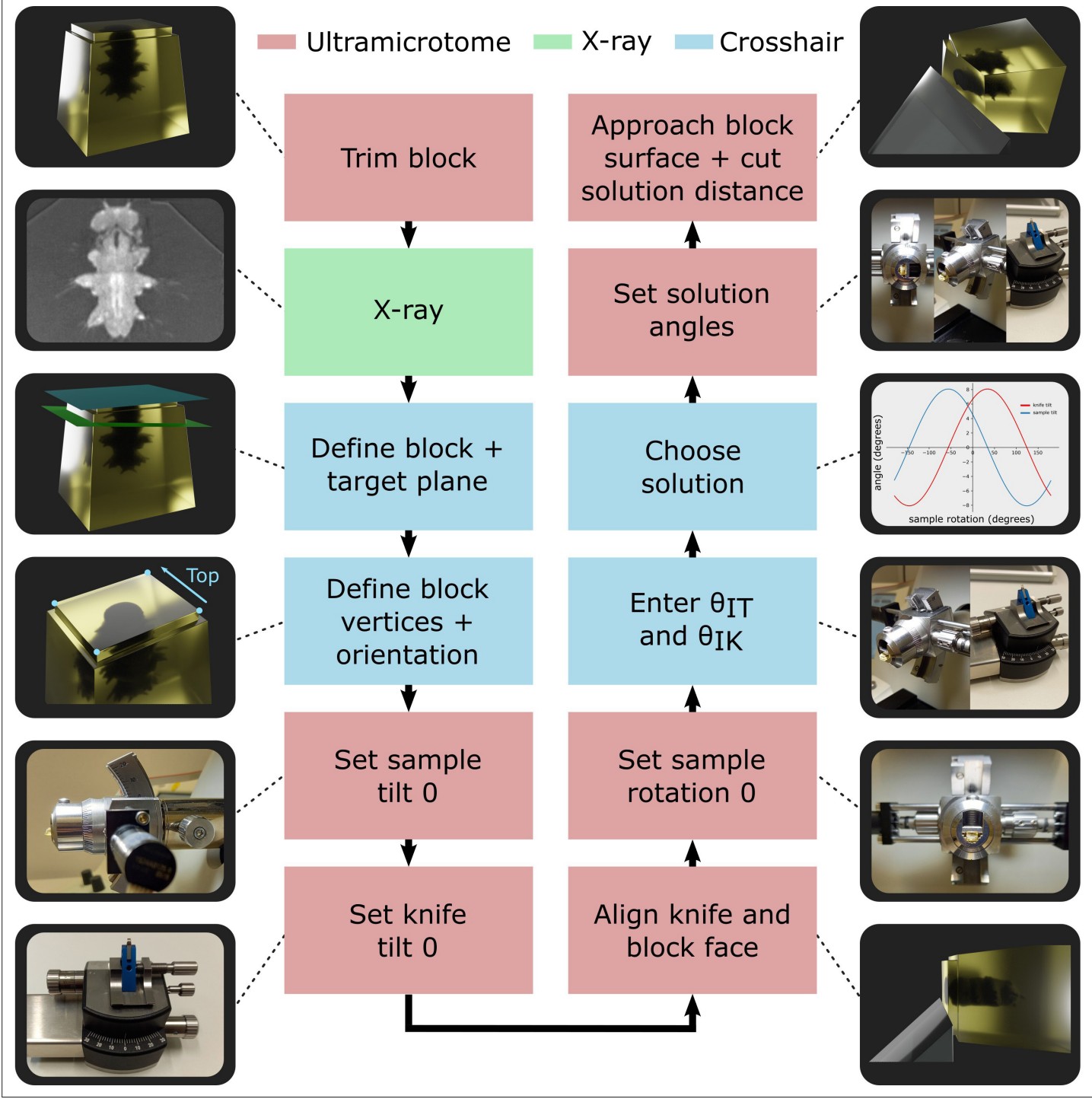

**Figure 4.** Targeting workflow. Summary of targeting workflow steps with representative images/3D renders.

There are a number of factors that limit the accuracy of the current system – the first is the resolution of the X-ray images. All X-ray imaging in this article was done with lab-based micro-CT systems, which offer a maximum resolution of around 1 micron isotropic voxel size. As all measurements are based on these images, this voxel size limits the achievable accuracy and contributes to our registration uncertainty. A future direction would be to combine Crosshair with higher-resolution X-ray imaging from a synchrotron. Synchrotrons are extremely powerful sources of X-rays and can provide resolutions in the range of hundreds or even tens of nanometres. In recent years, there have been a

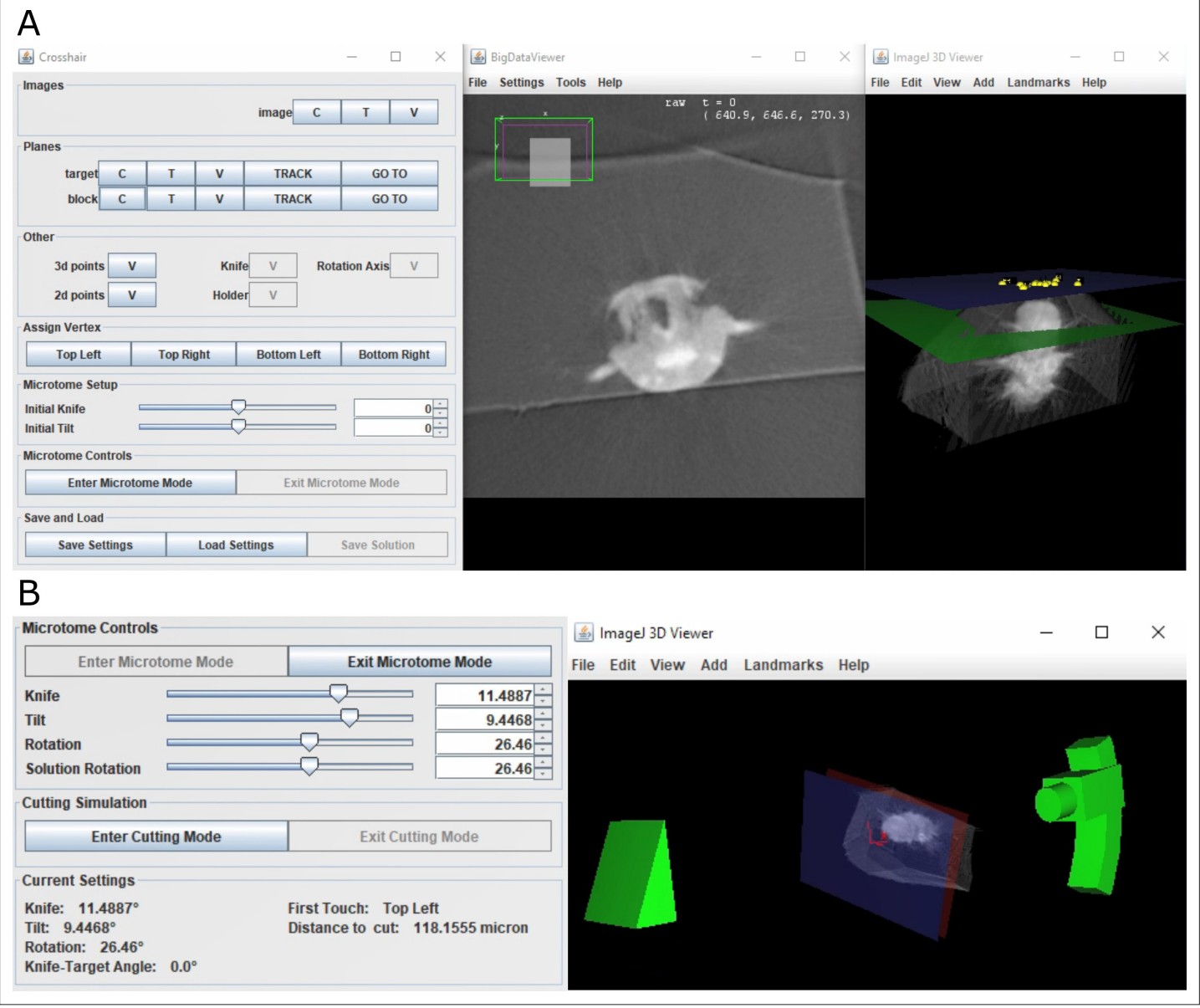

**Figure 5.** Crosshair. (**A**) Crosshair plugin user interface, with an example *Platynereis dumerilii* X-ray image. Left: controls; middle: BigDataViewer window showing 2D cross-section. Right: ImageJ 3D viewer showing volume rendering with block and target planes (blue and green, respectively). The green target plane is the same as the 2D slice shown. (**B**) Left: microtome mode controls and solution display. Right: corresponding 3D representation of ultramicrotome and sample.

The online version of this article includes the following figure supplement(s) for figure 5:

**Figure supplement 1.** Crosshair user interface.

number of studies that combine X-ray imaging from a synchrotron with EM (*Bosch et al., 2022*; *Kuan et al., 2020*; *Musser et al., 2021*), demonstrating the power of combining the large field of view and fast scan times of X-ray, with the high resolution of targeted EM acquisition. As this combination becomes ever more popular, it is more important than ever to have user-friendly targeting solutions like Crosshair.

Another accuracy limitation are the manual steps that remain in this targeting workflow. While we automate the motorised movements and angle/distance calculations, a number of steps still require manual use of the ultramicrotome, and will therefore vary in accuracy depending on how experienced the user is. For example, the setting of the zero positions of the knife and sample tilt, alignment of

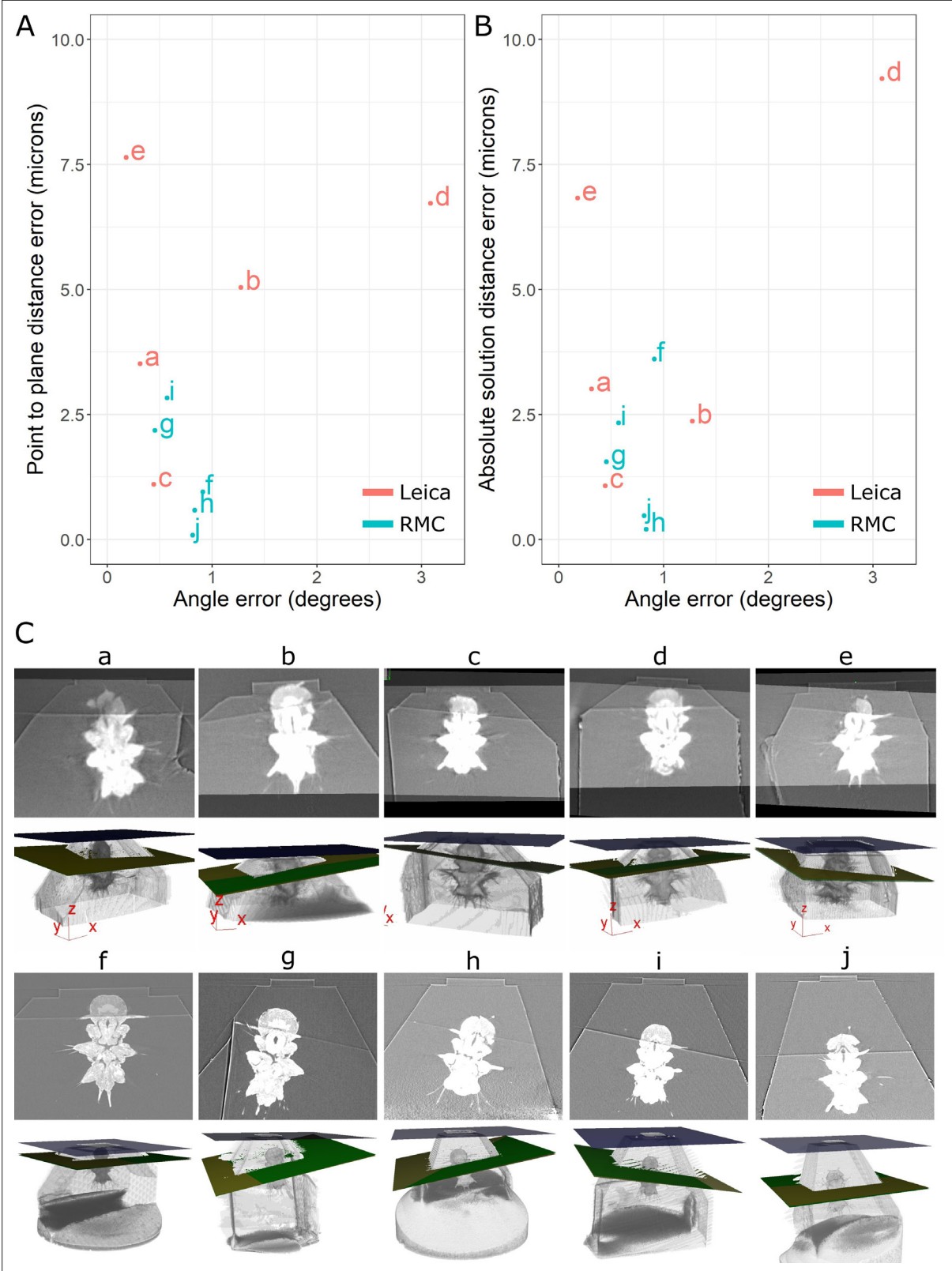

**Figure 6.** Accuracy test results. (**A**) Graph of point-to-plane distance error vs. angle error for all 10 samples (**a–j**). (**B**) Graph of absolute solution distance error vs. angle error. (**C**) For each sample (**a–j**), two images are shown. Top: registered before and after X-ray images. Bottom: snapshot of the 3D viewer in Crosshair. In the 3D view, the block surface is shown in blue, the target in green, and the plane reached in yellow. In most cases, the green and yellow planes are too close together to distinguish well. The axes shown in some of these images are the 3D viewer coordinate system.

*Figure 6 continued on next page*

*Figure 6 continued*

The online version of this article includes the following figure supplement(s) for figure 6:

**Figure supplement 1.** Registration accuracy.

**Figure supplement 2.** Sample mounting for trimming, X-ray imaging, and targeting for samples f–j.

the knife and block face, and approach of the knife to the block surface are all done manually. This will therefore introduce some variation due to human error.

Each of these manual steps should be minimised or eliminated in future work to improve accuracy. For example, the zero for the sample tilt could be set by precise measurement of the arc piece, rather than by eye. Also, it would be useful to find a more automated way to set the knife zero (although this is more complex as the knife is removed and replaced each time, and different knives are used for different purposes).

The manual alignment of the knife and block face could be removed if another method was found to determine the relation between the block surface and the sample holder. For example, the sample could be X-ray imaged directly inside the sample holder, although this is challenging in many micro-CT systems due to its large size. Alternately, a thin pin could be designed that would fit in the micro-CT and also fit tightly inside the sample holder at a fixed orientation. This is very similar to how samples f–j were mounted for targeting (see section 'Initial sample trimming and mounting' and *Figure 6— figure supplement 2*). Each of these samples was mounted to a thin aluminium pin at the start of the targeting process, and all trimming steps completed with a Gatan 3View Rivet holder that allows the pin to slot tightly inside. To eliminate the manual alignment step, this pin/holder would have to be adapted so the pin only slots inside in a fixed orientation. For example, by adding a groove to the side of the pin, that fits a corresponding projection on the sample holder. This would allow the relation between the block surface and sample holder to be calculated directly from the X-ray images alone.

The final manual step is the approach of the knife to the block surface. Currently, the user must move the knife forward and assess whether it has reached the block surface visually by examining the block through the ultramicrotome's binocular microscope. If cutting starts from one corner, it is easy to cut some distance before debris are visually detected on the knife (especially for less experienced ultramicrotome users). This results in errors in the final distance cut. This error increases each time the ultramicrotome is reset, meaning the greatest accuracy is achieved when the cutting distance is less than one feed length (200 microns). This being said, we did still achieve good accuracy with samples h, i, and j in this study that had cutting distances greater than one feed length (requiring one reset mid-run) (*Figure 6*). Even so, the requirement to manually reset and approach every 200 microns means that larger samples require more manual user intervention, and therefore more time and effort. Automation of this step is challenging and would require hardware improvements that extend the feed length of the ultramicrotome to minimise resets or a method to allow automatic resets. An automatic reset method would require a means to know the absolute distance between the knife and block surface, which is currently complex to achieve.

Moving away from improving the workflow's accuracy, it would also be interesting to expand the workflow's scope in future work. For example, Crosshair's calculations should be applicable to any 3D map (not only X-ray), and so could be expanded to targeting from 3D light microscopy. This would be useful for targeting fluorescently labelled structures during CLEM (correlative light and electron microscopy) as in *Ronchi et al., 2021*. Also, Crosshair could be expanded to allow multiple modalities to be used together; for example, displaying registered light microscopy and X-ray images of the same sample.

In addition, Crosshair could be expanded to automate hitting not only a target plane, but a specific location within that plane. For example, for the *Platynereis*, we may cut to a target plane that gives a cross-section through the entire head, and then automatically trim the block sides to leave only a certain sub-structure of interest. This is useful for methods such as FIB-SEM that cannot image large block surfaces. This would require precise, automated east–west movement of the knife (as already implemented on the RMC system), and expansion of the targeting calculations and Crosshair plugin.

In summary, this targeting workflow offers a first step towards fully automated targeting of structures within resin blocks for EM. By making it simpler, and faster, to target a specific plane of interest, we can achieve higher throughput and shorter overall acquisition times. Also, by providing a user-friendly

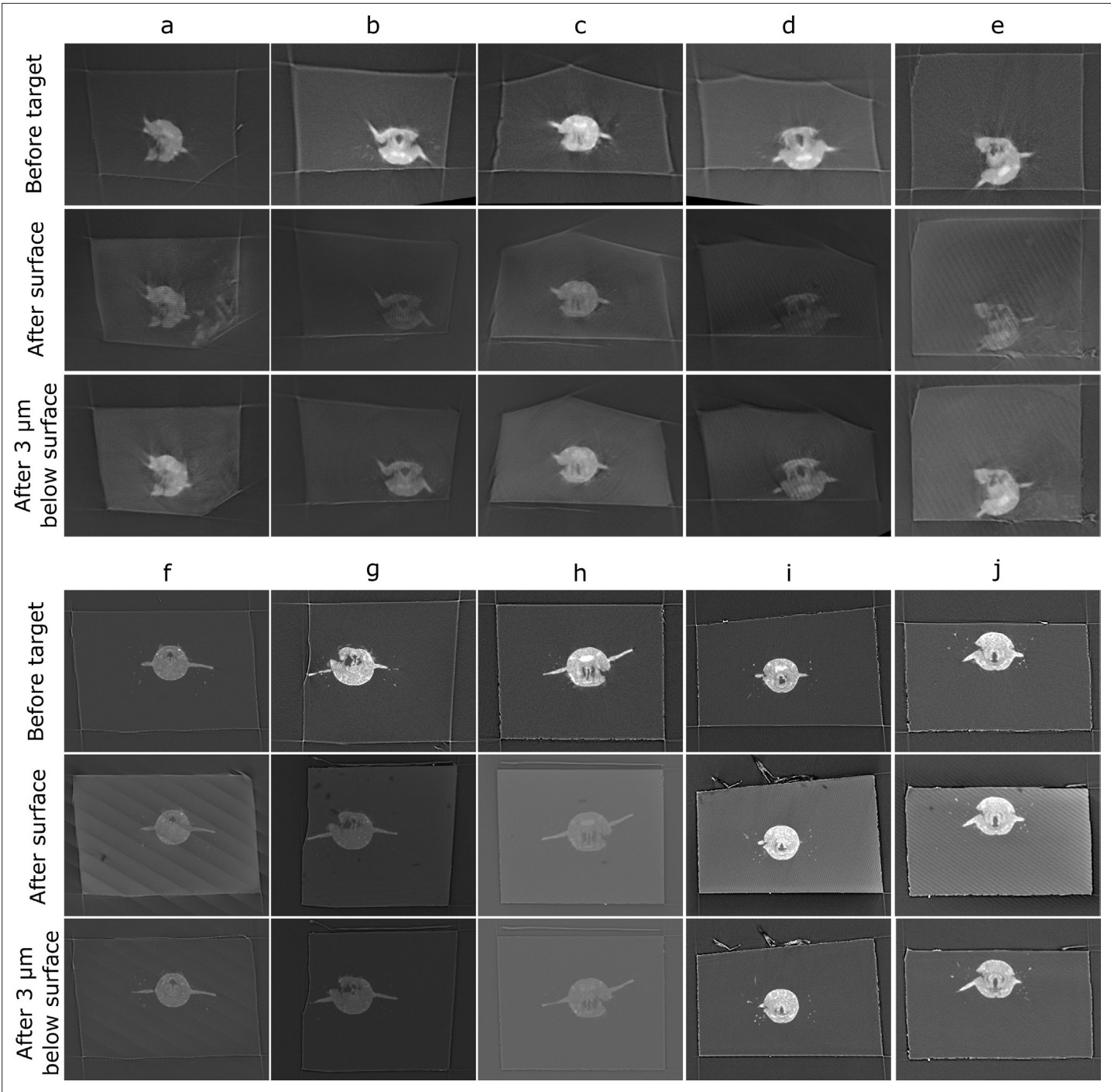

**Figure 7.** Comparison of target plane (from the before scan) to the block surface (after scan). Each sample (**a–j**) has a column with three images. Top row: the target plane from the before targeting scan. Middle row: the block surface from the after targeting scan. Bottom row: cross-section parallel to block surface, but 3 μm inside the block. The middle row images can be quite dark/noisy as they slice right along the surface of the block, so the bottom row is provided so the cross-section can be seen more easily.

The online version of this article includes the following figure supplement(s) for figure 7:

**Figure supplement 1.** Comparison of example *Platynereis* X-ray cross-sections, both acquired with 1 micron isotropic voxel size.

software for this process, we lower the barrier for newcomers to get started with 3D targeting. By providing the motorised ultramicrotome plans and software fully open-source, we hope that this work will enable more EM labs to rapidly target regions of interest for EM.

## Materials and methods
### Data availability

All X-ray images are available on EMPIAR with id 'EMPIAR-11055' (https://www.ebi.ac.uk/empiar/EMPIAR-11055/). All files for the accuracy measures tests (Crosshair settings and solutions, accuracy measures, registration metadata, registration accuracy measures) and calibration of the angular rotation of the ultramicrotomes are available on BioStudies with id 'S-BSST845' (https://www.ebi.ac.uk/biostudies/studies/S-BSST845). Diagrams, CAD files, and software for the two ultramicrotome systems are available in the following GitHub repositories: https://github.com/automated-ultramicrotomy/embl-system (*Meechan, 2022a*) and https://github.com/automated-ultramicrotomy/crick-system (*Jones, 2022*). The Crosshair Fiji plugin is available with tutorials and documentation at the following GitHub repository: https://github.com/automated-ultramicrotomy/crosshair (*Meechan, 2022b*), along with a video tutorial on YouTube for the Leica system (https://www.youtube.com/watch?v=OM712ECthGg).

### Ultramicrotome systems and calibration
#### Leica system

The motorised Leica system was designed as a simple add-on to any existing Leica UC7 (or similar ultramicrotome). This means it can easily be added/removed when it is required. Only one modification was made to the ultramicrotome itself – a hole was drilled into the cutting arm at a precisely vertical position (*Figure 1—figure supplement 1A*). This fits a small pin that was added to the back of the arc piece (*Figure 1—figure supplement 1B*) to keep the arc piece precisely vertical.

A new sample holder/arc and knife holder were purchased from Leica and modified with FAULHABER DC motors for each axis (*Figure 1B and D*). These motors have their own driver module that uses serial communication RS232 (12 V). A cable with a converter (RS232 to UART) is used to communicate with a Raspberry Pi running a Node-RED (https://nodered.org/) program for user interface and motor control. The Raspberry Pi is connected to a Raspberry Pi 7-inch touchscreen display and a StromPi3 battery. The battery is designed to hold up the shutdown process for 1 s and allow Linux to shut down properly, avoiding damage to the SD memory card that holds the operating system. The setup uses a desktop power supply (that provides 12 V) and connects to the StromPi3 battery module. The StromPi3 then provides the 5 V needed for the Raspberry Pi. All electronics are housed in a case with an adjustable arm for good screen visibility/ease of use. This allows the motor positions to be controlled and monitored easily by interacting with the touchscreen. The Node-RED user interface 'flow' json is available at https://github.com/automated-ultramicrotomy/embl-system/blob/main/node-red/node-red-flow.json (*Meechan, 2022a*). Once Node-RED is installed, the flow can be loaded by selecting 'import' from the menu and providing this .json file. This also requires the following nodes to be installed: node-red-dashboard (https://flows.nodered.org/node/node-red-dashboard) and node-red-node-serialport (https://flows.nodered.org/node/node-red-node-serialport).

Diagrams, CAD files, and software for this system are available on the following GitHub repository: https://github.com/automated-ultramicrotomy/embl-system (*Meechan, 2022a*).

#### RMC system

To motorise the existing RMC PTPC ultramicrotome, we replaced the manual operational knobs with Trinamic stepper motors and KHK gear sets. For each gear set, one gear was mounted on the shaft originally holding the knob, and another gear was mounted onto the stepper motor. There are five stepper motors and gear sets used in total to motorise the sample tilt/rotation on the sample holder, knife tilt, and stage NS/EW translations. Each stepper motor was chosen based on the consideration of both its maximum torque and its weight. For example, the motor to tilt the sample on the sample holder needs to be reasonably powerful as it is holding both the motor to rotate the sample and itself, which means it cannot be too heavy at the same time. The motor mount on the sample holder was

**Table 1.** Symbols for orientation calculation.

| Variable | Explanation |
| --- | --- |
| $\boldsymbol{R_x}$ | Rotation matrix about the x-axis |
| $\boldsymbol{R_y}$ | Rotation matrix about the y-axis |
| $\boldsymbol{R_z}$ | Rotation matrix about the z-axis |
| $\theta_T$ | Tilt angle of sample |
| $\theta_R$ | Rotation angle of sample |
| $\theta_K$ | Tilt angle of knife |
| $\theta_{IT}$ | Initial tilt of sample (on alignment) |
| $\theta_{IK}$ | Initial tilt of knife (on alignment) |
| $\theta_{to}$ | Target offset angle, i.e. block face to target plane z-axis rotation (measured from X-ray) |
| $\theta_{tr}$ | Target rotation angle, i.e. block face to target plane x-axis rotation (measured from X-ray) |

3D printed to reduce the weight and other motor mounts were manufactured in the Francis Crick Institute's mechanical workshop.

The retrofitted five motors were controlled by a Trinamic TMCM-6110 6-axis stepper motor driver board, which was linked via USB to a touchscreen PC. The GUI, as shown in *Figure 1—figure supplement 1D*, was developed with LabVIEW and installed onto this PC, allowing users to control the motions via keyboard/mouse, touchscreen, or even a joystick. The stepper motors can provide 51,200 microsteps per full 360° revolutions, which is equivalent to an accuracy of up to 0.00703125° per microstep.

Diagrams, CAD files, and software for this system are available on the following GitHub repository: https://github.com/automated-ultramicrotomy/crick-system (*Jones, 2022*).

## Calibration of Leica and RMC system

In order to calibrate the number of degrees rotated by these stepper motors via different gear sets (not only gear sets for retrofitted motors but also internal gears in the original RMC and Leica sample holder and knife holder to conduct the rotation), therefore, different gear ratios, we designed special adaptors to convert the rotational movement of the sample holder and the knife holder to the rotation of the dial of a Thorlabs CRM1PT/M rotation mount, whose Vernier scale provides 5 arcmin (0.083°) resolution. We rotated/tilted the sample and knife clockwise and anticlockwise travelling the whole range, then recorded the microstep numbers and readings from the rotation mount with roughly equal

**Table 2.** Constants for orientation calculation.

| Constant | Full expression |
| --- | --- |
| $A$ | $\cos(\theta_{IK} + \theta_{to})$ |
| $B$ | $\sin(\theta_{tr})\sin(\theta_{IK} + \theta_{to})$ |
| $C$ | $\sin(\theta_{IT})\sin(\theta_{IK} + \theta_{to})$ |
| $D$ | $\cos(\theta_{IT})\sin(\theta_{IK} + \theta_{to})$ |
| $E$ | $\cos(\theta_{tr})\sin(\theta_{IK} + \theta_{to})$ |
| $F$ | $\sin(\theta_{IT})\cos(\theta_{tr})$ |
| $G$ | $\sin(\theta_{tr})\cos(\theta_{IT})$ |
| $H$ | $\sin(\theta_{IT})\sin(\theta_{tr})$ |
| $I$ | $\cos(\theta_{IT})\cos(\theta_{tr})$ |

intervals. A linear regression was performed to finally work out the microstep numbers of the angle resolution, which we set to 0.1° to match the Vernier scale used (*Figure 1—figure supplements 2–4*).

Diagrams/CAD files of the calibration setup are available in the following GitHub repositories: https://github.com/automated-ultramicrotomy/embl-system (*Meechan, 2022a*) and https://github.com/automated-ultramicrotomy/crick-system (*Jones, 2022*). The tables of values used for the angular calibration are available on BioStudies (https://www.ebi.ac.uk/biostudies/studies/S-BSST845).

## Orientation calculation

For this section, we use the symbols in *Table 1* and the constants in *Table 2*. See *Figure 3B* for details of how $\theta_{to}$ and $\theta_{tr}$ are calculated.

## Forward kinematics equation

The forward kinematics equation was constructed by calculating the rotation matrices that relate each of the individual coordinate frames (*Figure 3A and B*, *Figure 3—figure supplement 1*). The World coordinate frame was fixed and defined x as parallel to the east–west movement of the ultramicrotome, y as parallel to the north–south movement, and z as vertical. The other coordinate frames could be imagined as attached to their respective components, moving with them as they move. These relations are summarised in the pose diagram in *Figure 3C*. We only focused on the orientation, and not relative position, as the position was solved in a separate process (see section 'Distance calculation'). When the knife and block surface were aligned, the orientation of the knife coordinate frame was the same as the block coordinate frame. This allowed the rotation matrix between the holder and block to be calculated by going the 'long way round' in this diagram, that is, from Holder, to Arc, to World, to Knife, and then to Block (*Figure 3D*). This was made simpler by defining the sample rotation to be zero at this aligned position.

The full forward kinematics equation, describing the rotation from the World coordinate frame to the target plane, was then

$$\boldsymbol{F} = \boldsymbol{R_x}(\theta_T)\boldsymbol{R_y}(\theta_R)\boldsymbol{R_x}(-\theta_{IT})\boldsymbol{R_z}(\theta_{IK})\boldsymbol{R_z}(\theta_{to})\boldsymbol{R_x}(\theta_{tr}) \tag{1}$$

This equation allowed us to accurately describe the orientation of the target plane for any set of input angles (sample tilt, sample rotation, and knife tilt).

## Sample tilt solution

The next step was to calculate the required inverse kinematics equation. For us, the constraint to satisfy was that the target coordinate frame's y-axis (which was the same as the target plane's normal) must be perpendicular to the world's z-axis. This would mean that the plane was vertical and, as the knife cuts vertically, could be reached by the knife (as long as it was within the angle range it could reach).

This was framed mathematically as

$$(\boldsymbol{F}\boldsymbol{y}) \cdot z = 0$$

where $\boldsymbol{F}$ is the forward kinematics equation defined above, $\boldsymbol{y}$ is the y vector $\begin{pmatrix} 0 \\ 1 \\ 0 \end{pmatrix}$, $z$ is the z vector $\begin{pmatrix} 0 \\ 0 \\ 1 \end{pmatrix}$, and · denotes the dot product. $\boldsymbol{F}\boldsymbol{y}$ gave the vector for the target plane y-axis in terms of the World coordinate frame. Constraining this so its dot product with the world z-axis must be 0 meant it must be perpendicular to world z, that is, the plane must be vertical.

Using the SymPy (*Meurer et al., 2017*) Python package to help with rearrangement gave

$$\begin{aligned} & \sin(\theta_T)\Big( \sin(\theta_{tr})\sin(\theta_{IT}) + \cos(\theta_{IT})\cos(\theta_{tr})\cos(\theta_{IK}+\theta_{to})\Big) \\ & + \cos(\theta_R)\cos(\theta_T)\Big( -\sin(\theta_{IT})\cos(\theta_{tr})\cos(\theta_{IK}+\theta_{to}) + \sin(\theta_{tr})\cos(\theta_{IT})\Big) \\ & + \sin(\theta_R)\cos(\theta_T)\Big( \sin(\theta_{IK}+\theta_{to})\cos(\theta_{tr})\Big) = 0 \end{aligned}$$

As $\theta_{IT}$, $\theta_{IK}$, $\theta_{to}$, and $\theta_{tr}$ were constants for a particular targeting run, we could replace these with the A–I constants defined above to make the expression easier to work with:

$$(AI + H)\sin(\theta_T) + (-AF + G)\cos(\theta_R)\cos(\theta_T) + E\sin(\theta_R)\cos(\theta_T) = 0$$

$$(-AF + G)\cos(\theta_R)\cos(\theta_T) + E\sin(\theta_R)\cos(\theta_T) = -(AI + H)\sin(\theta_T)$$

$$(-AF + G)\cos(\theta_R) + E\sin(\theta_R) = \frac{-(AI + H)\sin(\theta_T)}{\cos(\theta_T)}$$

$$\tan(\theta_T) = \frac{-AF + G}{-AI - H}\cos(\theta_R) + \frac{E}{-AI - H}\sin(\theta_R)$$

This meant that the final solution for the sample tilt was

$$\theta_T = \arctan\left(\frac{-AF + G}{-AI - H}\cos(\theta_R) + \frac{E}{-AI - H}\sin(\theta_R)\right) \tag{2}$$

This solution could of course be simplified further by combining these constants together to give something of the form

$$\theta_T = \arctan\left(C_1\cos(\theta_R) + C_2\sin(\theta_R)\right)$$

In practice, we used the form with constants A–I as this allowed the same constants to be used for this and the knife solution below.

The script for the SymPy parts is available in the Crosshair GitHub repository: https://github.com/automated-ultramicrotomy/crosshair/blob/master/python_scripts/multiple_solutions.py.

## Knife tilt solution

Now we had to find the knife angle that corresponded to a particular sample rotation. For this, we needed the signed angle between global y (i.e. ultramicrotome NS) and local y (the target plane's normal) in the z plane (i.e. the plane with the global z-axis as its normal). This was because the knife could only rotate about the z-axis, so we only needed the angle within this plane.

This was equal to

$$\theta_K = \arctan\left(\frac{(\mathbf{y} \times (\mathbf{F}\mathbf{y})) \cdot \mathbf{z}}{\mathbf{y} \cdot (\mathbf{F}\mathbf{y})}\right)$$

where $\mathbf{y}$ is the y vector $\begin{pmatrix} 0 \\ 1 \\ 0 \end{pmatrix}$, $\mathbf{z}$ is the z vector $\begin{pmatrix} 0 \\ 0 \\ 1 \end{pmatrix}$, $\cdot$ denotes the dot product, $\times$ denotes the cross-product, and $\mathbf{F}$ is the matrix from the forward kinematics equation defined in **Equation 1**.

Using SymPy (**Meurer et al., 2017**) to help with rearrangement, and substitution of the same constants A–I from above, this became

$$\theta_K = \arctan\left(\frac{E\cos(\theta_R) + (AF - G)\sin(\theta_R)}{-E\sin(\theta_R)\sin(\theta_T) + (AF - G)\sin(\theta_T)\cos(\theta_R) + (AI + H)\cos(\theta_T)}\right)$$

This equation gives the corresponding signed knife angle for all values of $\theta_T$ and $\theta_R$, but we only wanted the values that are valid given the solution calculated above for the sample tilt. That is, we only wanted knife angles where the target plane was vertical, and therefore reachable by the knife. Therefore, we substituted in the solution from above (**Equation 2**) for $\theta_T$.

This gave (with some rearrangement/simplification with SymPy)

$$\theta_K = \arctan\left(\frac{\left(AI + H\right)\left(E\cos(\theta_R) + (AF - G)\sin(\theta_R)\right)}{\left(\sqrt[2]{\left(AI + H\right)^2 + \left(E\sin(\theta_R) + (-AF + G)\cos(\theta_R)\right)^2}\right)|AI + H|}\right) \tag{3}$$

This solution could of course be simplified further by combining these constants together to give something of the form

$$\theta_K = \arctan\left(\frac{C_3\left(C_4\cos(\theta_R) + C_5\sin(\theta_R)\right)}{\left(\sqrt[2]{C_3{}^2 + \left(C_4\sin(\theta_R) - C_5\cos(\theta_R)\right)^2}\right)|C_3|}\right)$$

In practice, we used the form with constants A–I in the Crosshair code.

Therefore, with *Equation 2* and *Equation 3*, we could find the corresponding sample tilt and knife tilt for any given sample rotation.

The script for the SymPy parts is available in the Crosshair GitHub repository: https://github.com/automated-ultramicrotomy/crosshair/blob/master/python_scripts/multiple_solutions.py (*Meechan, 2022b*).

## Distance calculation

Once the orientation was solved, we calculated the distance to cut from the sample to reach the target plane. As the cutting direction (NS) may not have been parallel to the target plane normal, this had to be done in two steps. First, the perpendicular distance was calculated from the target plane to the furthest point of the block face, then this was compensated for the offset between this and NS.

When the orientation was solved, we knew that the target plane must be vertical. Therefore, to find the NS distance we only needed to compensate for the current knife angle (*Figure 3—figure supplement 2*).

This meant that the distance was

$$D_{NS} = \frac{D_P}{\cos(\theta_K)}$$

where $D_{NS}$ is the NS distance, $D_P$ is the perpendicular distance between the target plane and the furthest surface point, and $\theta_K$ is the knife angle.

## Crosshair

Crosshair is a Fiji (*Schindelin et al., 2012*) plugin written in Java and is freely available from the following GitHub repository: https://github.com/automated-ultramicrotomy/crosshair; *Meechan, 2022b*. A user guide is also available (https://github.com/automated-ultramicrotomy/crosshair/wiki; *Meechan, 2022b*), along with a video tutorial of the entire workflow demonstrated on the Leica system (https://www.youtube.com/watch?v=OM712ECthGg). It can also be easily installed via a Fiji update site (instructions in the GitHub repository).

It builds on top of BigDataViewer (*Pietzsch et al., 2015*) for viewing images in 2D, and the ImageJ 3D Viewer (*Schmid et al., 2010*) for viewing images in 3D. It also builds on a number of functions from Christian Tischer's imagej-utils repository (https://github.com/embl-cba/imagej-utils; *Tischer et al., 2022a*).

## RegistrationTree

RegistrationTree is a Fiji (*Schindelin et al., 2012*) plugin written in Java and is freely available from the following GitHub repository: https://github.com/K-Meech/RegistrationTree; *Meechan, 2022c*. A user guide is also available (https://github.com/K-Meech/RegistrationTree/wiki; *Meechan, 2022c*). It can be easily installed via a Fiji update site (instructions in the GitHub repository).

RegistrationTree is a wrapper around BigWarp (*Bogovic et al., 2016*) and elastix (*Klein et al., 2010*; *Shamonin et al., 2013*) registration software. It allows easy passing of images between the two software tools, as well as easing the integration of large images with elastix. Users can build arbitrary trees of affine transformations to register their images together and visualise/compare these easily. RegistrationTree uses BigDataViewer (*Pietzsch et al., 2015*) for 2D visualisation, and BigDataViewer style file formats (HDF5 and N5) to allow very large images to be viewed efficiently on a normal laptop. It builds on top of Christian Tischer's elastixWrapper (*Tischer, 2019*, https://github.com/embl-cba/elastixWrapper) and imagej-utils repository (https://github.com/embl-cba/imagej-utils; *Tischer et al., 2022a*). Also, the open-source repositories for image-transform-converters (https://github.com/image-transform-converters/image-transform-converters; *Tischer et al., 2022b*) and bigdataviewer-playground (https://github.com/bigdataviewer/bigdataviewer-playground; *Chiaruttini et al., 2022*).

## Targeting workflow

The targeting workflow is summarised in *Figure 4*. We expand on those steps in detail here in the form of a step-by-step protocol. They are also described in detail in the Crosshair user guide (https://github.com/automated-ultramicrotomy/crosshair/wiki; *Meechan, 2022b*) and in a video tutorial

demonstrated on the Leica system (https://www.youtube.com/watch?v=OM712ECthGg). Note that there are some differences between the motorised Leica and RMC systems that are highlighted throughout the workflow.

### Trim block

Trim the block at the ultramicrotome with a diamond knife. A flat surface must be made that has four corners and straight lines in between. Make sure the sides of the block face are trimmed deep enough, so the knife will always touch part of this face first, and not other parts of the block (even if the solution requires a more extreme set of angles). We usually trim about 40–100 microns, depending on the sample.

Ideally, the block surface should be within one ultramicrotome feed length of the target for the greatest accuracy (i.e. 200 microns or less for both the Leica and RMC systems). Every time the cutting feed is reset, the approach of the knife to the block must be repeated, and some accuracy will be lost. If necessary, the distance can be greater though; for example, samples h–j in this study had cutting depths greater than one feed length (requiring one reset mid-run), and their accuracy was not greatly affected (see *Figure 6*).

Note that there are a number of different ways to mount the sample for trimming and later X-ray imaging. For example, see the different solutions we used in the section 'Initial sample trimming and mounting'.

### X-ray

Convert the X-ray image to 8-bit (if it is not already), for example, in Fiji with Image – Type – 8-bit. The image can also be cropped at this stage if there is a lot of empty area outside the resin block. Open the X-ray in Crosshair and compare the 3D view/image stack and the sample. If it appears as a mirror image of the sample, then flip the image stack. This can be done, for example, in Fiji with Image – Transform – Flip Vertically, or Image – Transform – Flip Z.

### Define block + target plane

Open the X-ray image in Crosshair and set the block and target planes. The target plane can be set using the 'TRACK' button and navigating in the 2D viewer. The block face can be set by placing a series of points on the block surface and fitting a plane to them.

### Define block vertices + orientation

Place one point (vertex) on each of the four corners of the block face. Then, decide on the orientation the block will have in the ultramicrotome, that is, which side of the block will face up? Mark the points accordingly as 'Top Left', 'Top Right', 'Bottom Left', and 'Bottom Right' in Crosshair.

Once this is complete, it is good practice to 'Save Settings' in Crosshair to make a .json file as a record of the planes and points.

### Set sample tilt 0

Set the sample tilt to zero by eye (just by looking at the scale on the arc piece and trying to get it as close to zero as possible). For the Leica system, press 'Set Zero' in the 'sample tilt' page of the ultra-microtome user interface. For the RMC system, make a note of the current sample tilt angle.

### Set knife tilt 0

For the knife zero, there is a slightly more complicated procedure than for the sample tilt. We assumed that since the knife is removed and replaced every time, there may be more variation in its position and therefore an extra step may be required to accurately find its proper zero.

Put a blank block into the ultramicrotome sample holder (i.e. just resin with no sample). Set the knife tilt to zero by eye (just by looking at the scale on the knife holder and trying to get it as close to zero as possible). Cut into the blank block at this orientation to make a flat face. Back the knife away and rotate the sample holder exactly 90° (using the motorised ultramicrotome controls). Align the knife to this face again (only adjusting the knife tilt!). For the Leica system, press 'Set Zero' in the

'knife tilt' page of the ultramicrotome user interface. For the RMC system, make a note of the current knife tilt angle.

## Align knife and block face

Replace the blank block with the resin-embedded sample. Make sure it is placed the same way up as indicated in Crosshair.

Align the knife and block face, ensuring the bottom edge of the block is also aligned along the cutting edge of the knife. Any axis can be freely changed for this step, just as in normal ultramicrotome usage. For the Leica system, the motors can be disabled for this step, and the alignment done manually as with normal ultramicrotome usage. For the RMC system, the motors cannot be disabled, and the motors must be used for this alignment.

## Set sample rotation 0

For the Leica system, set the sample rotation to zero by pressing 'Set Zero' in the 'sample rotation' page of the ultramicrotome user interface. For the RMC system, make a note of the current sample rotation angle.

## Enter $\theta_{IK}$ and $\theta_{IT}$

Enter the current knife angle and sample tilt angle into Crosshair. For the RMC system, make sure you adjust for the recorded 'zero' position, for example, if the current angle is 10°, and the zero was 2°, then you would enter 8° into Crosshair.

## Choose solution

Use the Crosshair 'Microtome Mode' to cycle through all the possible solutions, and choose one to use. The target plane will turn red in the 3D view when there is a valid solution.

Once this is complete, it is good practice to 'Save Solution' in Crosshair to make a .json file as a record of the solution.

**Table 3.** Table of *Platynereis* preparation steps.

| Step no. | Step | Time | Temperature | Microwave (W) | Vacuum |
|---|---|---|---|---|---|
| 1 | 2% paraformaldehyde, 2.5% glutaraldehyde in seawater | 2 min | 21°C | 100 | On |
| 2 | 2% paraformaldehyde, 2.5% glutaraldehyde in 0.1 M cacodylate | 2 × 14 min (2 min on/off cycles) | 21°C | 100 | On |
| 3 | 0.1 M cacodylate | 1 immediate Then 2 × 40 s. | 21°C | 100 | On |
| 4 | 2% OsO$_4$ in 0.1 M cacodylate | 2 × 14 min (2 min on/off cycles) | 21°C | 100 | On |
| 5 | 2.5% K$_4$[Fe(CN)$_6$].3H$_2$O in 0.1 M cacodylate | 2 × 14 min (2 min on/off cycles) | 21°C | 100 | On |
| 6 | Water wash | 1 immediate Then 2 × 40 s | 21°C | 100 | On |
| 7 | 1% TCH unbuffered | 2 × 14 min (2 min on/off cycles) | 40°C | 100 | On |
| 8 | Water wash | 1 immediate Then 2 × 40 s | 40°C | 100 | On |
| 9 | 2% OsO$_4$ aqueous | 2 × 14 min (2 min on/off cycles) | 21°C | 100 | On |
| 10 | Water wash | 1 immediate Then 2 × 40 s | 21°C | 100 | On |
| 11 | Dehydration series in ethanol (25%, 50%, 75%, 3 × 100%) | 40 s each | 10°C | 250 | Off |
| 12 | Infiltration series in Durcupan (25%, 50%, 75%, 90%, 3 × 100%) | 3 min each | 21°C | 150 | On |
| 13 | 100% Durcupan | Overnight | 21°C | N/A | N/A |
| 14 | Polymerisation in oven | 48 hr | 60°C | N/A | N/A |

## Set solution angles

Back the knife away from the block to ensure there is no risk of hitting it. Using the motors, move each axis to the angles stated in the solution from Crosshair. For the RMC system, make sure you adjust for the recorded 'zero' position of each axis, for example, if the solution angle is 10°, and the zero was 2°, then you would move the ultramicrotome to 12° for that axis.

## Approach block surface + cut solution distance

Manually approach the block and get as close as possible to its surface. Any error in this distance (i.e. starting before touching the block, or once it has already been cut into) will contribute to the final error. A good solution is to get as close as possible with the ultramicrotome's NS movement, then set the cutting thickness to some small value (e.g. 70–100 nm). Then cutting can be started slowly, carefully watching the knife edge through the ultramicrotome's binocular at high magnification, and stopped as soon as any debris is visible on the knife/any pieces being cut from the block. This ensures the knife is as close as possible to the true surface.

Once complete, cut the distance specified in the Crosshair solution.

## *Platynereis* sample preparation

*Platynereis* were prepared with a modification of the method from *Hua et al., 2015*, aided by microwave processing (see details in *Table 3*). *Platynereis* were flat embedded in thin pieces of resin.

## Accuracy tests

### Initial sample trimming and mounting

Samples were prepared as specified in the section '*Platynereis* sample preparation'. Individual *Platynereis* were then cut out in small pieces of resin with a razor blade.

For samples a–e, these pieces were glued to larger resin blocks to make them easier to handle in the ultramicrotome. These blocks were then inserted into standard Leica sample holders and mounted to the ultramicrotome for trimming.

For samples f–j, the small pieces of resin containing *Platynereis* were mounted onto aluminium pins using conductive epoxy glue (ITW Chemtronics). The pin was then inserted into a Gatan 3View Rivet Holder (Gatan 3VRHBM) and mounted to the ultramicrotome (*Figure 6—figure supplement 2A, B*) for trimming.

Blocks were trimmed close to the *Platynereis* with a diamond knife, so the block surface for most samples was less than one feed length (200 microns) from the target. A flat surface was trimmed with a rectangular shape as required for Crosshair. Note that samples h–j had cutting distances greater than one feed length, but their accuracy was not greatly affected (see *Figure 6*).

### X-ray imaging

Samples a–e were imaged with a Bruker SkyScan 1272, with an isotropic voxel size of 1 micron. The resin blocks were mounted onto small pins with wax for the X-ray, and then removed from these pins for all further steps at the ultramicrotome. The log files with the specific settings for each sample are provided on EMPIAR alongside the images (https://www.ebi.ac.uk/empiar/EMPIAR-11055/). Imaging each sample took 2 hr to complete. Fiji was then used to convert the X-ray images to 8-bit, flip them vertically (Image – Transform – Flip Vertically), and crop to remove empty areas outside the resin block.

Samples f–j were imaged with a Versa 510 X-ray microscope (Zeiss). Aluminium pins were mounted to a Versa 510 sample holder (*Figure 6—figure supplement 2C*) and imaged at a voltage of 40 kV, power of 3 W. No filter was used, and images were acquired at a binning of 2 with an exposure time of 10 s. Pixel size was chosen to be as close to 1 micron as possible (actual reading varied between 0.9999 and 1.0001 microns for different runs). The field of view thus varied between 1014.9 and 1015.1 microns$^2$. A total of 1601 projections were captured over the 360° tilt range. Imaging each sample took 5 hr and 30 min to complete. Fiji was then used to convert the X-ray images to 8-bit, adjust their brightness and contrast, flip vertically (Image – Transform – Flip Vertically), and finally reverse the image stack (Image – Stacks – Tools – Stack Sorter – Reverse).

For all samples, raw and reconstructed X-ray images are available (before and after targeting), as well as the X-ray images used for Crosshair (i.e. after any flipping, cropping, etc., was done) (https://www.ebi.ac.uk/empiar/EMPIAR-11055/).

## Crosshair and targeting

The X-ray images were then processed with Crosshair. The block face plane was set by fitting to a series of points placed on the surface. The target plane was set by browsing in 2D to find a plane that cut through both anterior dorsal cirrus that protrude from either side of the *Platynereis* head. The remaining targeting steps were completed as specified in the workflow in the section 'Targeting workflow'. For samples a–e, all steps were completed using a standard Leica sample holder. For samples f–j, a Gatan 3View Rivet Holder was used (as specified in the section 'Initial sample trimming and mounting').

All Crosshair settings and solution files are available on BioStudies (https://www.ebi.ac.uk/biostudies/studies/S-BSST845).

## Registration

After targeting, the samples were imaged with X-ray again as described in the section 'X-ray imaging'. The before and after X-ray images were then registered using our RegistrationTree plugin (see section 'RegistrationTree'). For this, the before scan (moving image) was registered to the after scan (fixed image). This used two registration steps: first, a rough point-based registration with BigWarp (*Bogovic et al., 2016*), followed by an intensity-based registration with elastix (*Klein et al., 2010*; *Shamonin et al., 2013*). Both steps were limited to rotation and translation only. Once complete, this was exported from RegistrationTree as a BigDataViewer xml file containing the required transform in the 'MOVING' space, that is, so it transformed the after scan onto the before scan.

For some of our X-ray images, we saw slight variation in the before and after scans of the same sample (particularly with the Bruker system). We therefore also calculated an approximate error in the accuracy of our registration. This was done by manually labelling four corresponding points in each registered pair of X-ray images for all samples. For example, the tips of the *Platynereis* parapodia (appendages used for swimming and crawling). These were placed using BigWarp's features for point placement, and then exported to csv files. The Euclidean distance between each pair of points was then calculated for *Figure 6—figure supplement 1*. This gave a mean error of 1.9 microns (for the points labelled repeat 1). As there was also some error due to manual placement of these points, we repeated the process twice for the same locations and images (repeats 1 and 2 in *Figure 6—figure supplement 1A*). The mean difference between these repeats was 1.1 microns. As the average difference between repeats was less than the error we measured, it implies that this likely comes from error in the registration, rather than inaccuracy of the manual point placement. As they are of a similar size though, it does emphasise the difficulty of accurately measuring these small distances, given that the voxel size of our micro-CT was only 1 micron.

For all samples, the RegistrationTree and registration accuracy files are available on BioStudies (https://www.ebi.ac.uk/biostudies/studies/S-BSST845).

## Accuracy measures

Accuracy measures were calculated using features built-in to the Crosshair Fiji plugin ('Measure Targeting Accuracy' command). This takes the before X-ray image, the registered after X-ray image and the Crosshair settings and solution files, and allows user-friendly calculation of accuracy measures. A plane was fitted to the after scan's block surface by manually placing a series of points on its surface (same workflow as fitting the planes in usual Crosshair usage). A point was also placed on the before scan's target plane approximately at the centre of the *Platynereis* cross-section. Then, the 'Save Measures' option was used to save a simple .json file containing the measures.

This included three measures: one for the angle error, one for the solution distance error, and one for the point-to-plane distance error. The angle error was calculated as the absolute angle between the normals of the target plane and the after scan surface. The distance error was more complex to calculate as any planes that are not exactly parallel will intersect at some location (i.e. distance = 0). Therefore, we provided two different measures for this: the solution and point-to-plane distance.

The solution distance calculated the approximate difference between the solution distance and the distance actually cut (inferred from the Crosshair settings, and after block surface position). It was therefore a measure of how accurately the solution's distance was cut, rather than a direct measure of how close the final surface was to the target plane. This was calculated by determining the shortest distance between the predicted first touch point (on the original block surface) and the after scan

block surface. This was then converted to the NS cutting distance by compensating for the solution knife angle:

$$D_{NS} = \frac{D_P}{\cos(\theta_K)}$$

where $D_{NS}$ is the NS distance, $D_P$ is the perpendicular distance between the after surface plane and the first touch point, and $\theta_K$ is the knife angle from the chosen solution.

The final distance error was then

$$\text{Error} = D_{NS} - D_{Sol}$$

where $D_{NS}$ is the NS distance, and $D_{Sol}$ is the distance from the chosen solution. A positive error indicates that the sample was trimmed too far, while a negative error indicates it was not trimmed enough.

Note that these calculations assumed that cutting started from the predicted point. If the angle was so far off that cutting started from another point, this measure would be erroneous. This is something that was checked at the start of each run though by examining the block and checking that the correct vertex was touched first. This calculation also assumed that the knife angle was the same as stated in the solution. If there was any error in the knife angle setting, then this would also affect this accuracy measure.

To have a more direct measure of how close the final surface was to the target plane, we also provided the point-to-plane distance error. This is calculated as the shortest distance between a point on the before target plane (near to the centre of the region of interest) and the after block surface. Note that therefore this measure will vary depending on where this point was placed within the target plane.

See the following website for the calculation scripts: https://github.com/automated-ultramicro-tomy/crosshair/blob/master/src/main/java/de/embl/schwab/crosshair/targetingaccuracy/Accuracy-Calculator.java (*Meechan, 2022b*). For all samples, all accuracy settings and accuracy measures files are available on BioStudies (https://www.ebi.ac.uk/biostudies/studies/S-BSST845).

## Acknowledgements

We thank the Electron Microscopy Core Facility (EMBL) for access to the instrumentation for electron microscopy and the Arendt lab (EMBL Heidelberg) for providing the *Platynereis dumerilii* samples. We also thank Christian Tischer (Centre for Bioimage Analysis, EMBL Heidelberg) for his help with creating the Fiji plugins, and the Mechanical Workshop at The Francis Crick Institute for their mechanical expertise and practical assistance during the development.

## Additional information

### Funding

| Funder | Grant reference number | Author |
|---|---|---|
| Francis Crick Institute | FC001999 | Wei Guan<br>Azumi Yoshimura<br>Christopher J Peddie<br>Martin L Jones<br>Lucy Collinson |
| European Molecular Biology Laboratory | Predoctoral Fellowship | Kimberly Meechan |

| Funder | Grant reference number | Author |
|---|---|---|
| European Molecular Biology Laboratory | | Alfons Riedinger<br>Vera Stankova<br>Rosa Pipitone<br>Arthur Milberger<br>Helmuth Schaar<br>Inés Romero-Brey<br>Rachel Templin<br>Nicole L Schieber<br>Yannick Schwab |

The funders had no role in study design, data collection and interpretation, or the decision to submit the work for publication.

## Author contributions

Kimberly Meechan, Conceptualization, Data curation, Software, Formal analysis, Validation, Investigation, Visualization, Methodology, Writing – original draft, Writing – review and editing; Wei Guan, Conceptualization, Data curation, Software, Formal analysis, Validation, Investigation, Visualization, Methodology, Writing – review and editing; Alfons Riedinger, Vera Stankova, Software, Methodology; Azumi Yoshimura, Formal analysis, Validation, Investigation, Visualization, Writing – review and editing; Rosa Pipitone, Inés Romero-Brey, Rachel Templin, Investigation; Arthur Milberger, Visualization, Methodology; Helmuth Schaar, Methodology; Christopher J Peddie, Conceptualization, Validation, Methodology, Writing – review and editing; Nicole L Schieber, Conceptualization, Supervision, Project administration; Martin L Jones, Conceptualization, Supervision, Project administration, Writing – review and editing; Lucy Collinson, Conceptualization, Resources, Formal analysis, Supervision, Funding acquisition, Validation, Investigation, Project administration, Writing – review and editing; Yannick Schwab, Conceptualization, Resources, Supervision, Funding acquisition, Project administration, Writing – review and editing

## Author ORCIDs

Kimberly Meechan (ID) http://orcid.org/0000-0003-4939-4170
Rosa Pipitone (ID) http://orcid.org/0000-0002-6855-5614
Rachel Templin (ID) http://orcid.org/0000-0003-2800-5826
Christopher J Peddie (ID) http://orcid.org/0000-0002-8329-5419
Nicole L Schieber (ID) http://orcid.org/0000-0002-3212-5057
Martin L Jones (ID) http://orcid.org/0000-0003-0994-5652
Yannick Schwab (ID) http://orcid.org/0000-0001-8027-1836

## Decision letter and Author response

Decision letter https://doi.org/10.7554/eLife.80899.sa1
Author response https://doi.org/10.7554/eLife.80899.sa2

# Additional files

## Supplementary files

• MDAR checklist

## Data availability

All X-ray images are available on EMPIAR with id 'EMPIAR-11055': https://www.ebi.ac.uk/empiar/EMPIAR-11055/. All files for the accuracy measures tests (Crosshair settings and solutions, accuracy measures, registration metadata, registration accuracy measures) and calibration of the angular rotation of the ultramicrotomes are available on BioStudies with id 'S-BSST845': https://ebi.ac.uk/biostudies/studies/S-BSST845. Diagrams, CAD files and software for the two ultramicrotome systems are available in the following github repositories: https://github.com/automated-ultramicrotomy/embl-system (copy archived at swh:1:rev:78471b53c3dc8b4f6d6fac357171c42cf5af8134) and https://github.com/automated-ultramicrotomy/crick-system (copy archived at swh:1:rev:d-7f91bf6dba69cfa823f8ebcef96da907d769a0e). The Crosshair Fiji plugin is available with tutorials and documentation at the following github repository: https://github.com/automated-ultramicrotomy/crosshair (copy archived at swh:1:rev:43291bc07467612e4561e402e623a9fad5ae77bc),

along with a video tutorial on YouTube for the Leica system: https://www.youtube.com/watch?v=OM712ECthGg.

The following datasets were generated:

| Author(s) | Year | Dataset title | Dataset URL | Database and Identifier |
|---|---|---|---|---|
| Meechan K, Guan W, Riedinger A, Stankova V, Yoshimura A, Pipitone R, Milberger A, Schaar H, Romero-Brey I, Templin R, Peddie CJ, Schieber NL, Jones ML, Collinson L, Schwab Y | 2022 | X-ray images | https://www.ebi.ac.uk/empiar/EMPIAR-11055/ | EMPIAR, EMPIAR-11055 |
| Meechan K, Guan W, Riedinger A, Stankova V, Yoshimura A, Pipitone R, Milberger A, Schaar H, Romero-Brey I, Templin R, Peddie CJ, Schieber NL, Jones ML | 2022 | Files for accuracy measures and ultramicrotome calibration | https://www.ebi.ac.uk/biostudies/studies/S-BSST845 | BioStudies, S-BSST845 |

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
