## [Editor Report]

Meechan et al. present a systematic approach to semi-automate an ultramicrotome operation for targeting a specific plane aided by X-ray tomography measurements, and it is a fundamental work of great interest to any users of electron microscopy (EM), particularly when targeting the imaging of thin sections in a select region of interest by ultramicrotomy, or when targeting volume EM of select sample regions. The article documents with exceptional detail a workflow including both microtome modifications and software adaptations for semi-automated targeting of structures with micrometre precision, resulting in a faster and more accurate orientation of the image acquisition planes for volume electron microscopy, a task that has traditionally been difficult and time-consuming. Therefore, this work will reduce sample preparation labour and, critically, facilitate the comparison of the ultrastructure of multiple samples. The method is based on X-ray imaging acquisition prior to any sectioning and proposes a solution for the two instruments commercially available in the field, and by transparently sharing all the data, hardware, and software, and describing every detail of the workflow, this fundamental method can be readily adopted by any practitioner, enabling its wide application – it is a key step in the field regarding speed-up, accuracy, and reproducibility in electron microscopy.

---

## [Decision Letter]

**Decision letter after peer review:**

Thank you for submitting your article "Crosshair, semi-automated targeting for electron microscopy with a motorised ultramicrotome" for consideration by *eLife*. Your article has been reviewed by 3 peer reviewers, and the evaluation has been overseen by a Reviewing Editor and Anna Akhmanova as the Senior Editor. The following individuals involved in the review of your submission have agreed to reveal their identity: Christel Genoud (Reviewer #1); Song Pang (Reviewer #2); Michaela Wilsch-Bräuninge (Reviewer #3).

Essential revisions:

1) Please note the suggested expansion of figure 3 to include its own supplements 1 and 2 into the main figure for clarity of exposition and to be able to logically follow the argument. And also to relabel the axes for clarity, for example, to move the axis caption to be near the axis itself, so that the text runs nearly flush, and parallel, with the axis it captions. Otherwise, it's unclear which label corresponds to which label.

2) Please discuss the limitations in the ultramicrotome feed length and how these can affect automation.

*Reviewer #1 (Recommendations for the authors):*

This article is providing a critical tool to all the community.

I cannot judge the geometry approach as I am not good enough in geometry to critically assess or propose any improvement in this part.

Figures are of outstanding quality and make the article easy to read and interesting even for readers not directly interested in implementing the approach.

I do not see at this point any need for major improvements or modifications of the article.

As the limitations are explained by the authors at the end and the future developments are proposed and openly discussed, I do not have suggestions that will improve this article sufficiently to delay its publication and make a second run of revision.

*Reviewer #2 (Recommendations for the authors):*

Enabling a solution to avoid the reset of the cutting feed (with a feed of 500 nm or less) will be a critical step towards automation. This feature will not only greatly reduce the labor requirement – freeing users from babysitting microtome during trimming, but also improve the target accuracy and operation throughput.

Figure 3 is difficult to follow without the prelude of Figure 3—figure supplements 1 and 2. Combining these two supplement figures into Figure 3 will be more intelligible. Additionally, the axes labels in Figures 3 C and 3 D are confusing. It would be helpful to spell out in the caption which label is for x and which is for y, respectively.

*Reviewer #3 (Recommendations for the authors):*

The comparison between two different setups for the motorization of the microtomes is a very useful piece of data. In the description, however, the difference between the 3 motors for the Leica system and the additional translational stage motors for the RMC is not explained. It may be useful to give a simple explanation.

The print version of the equations is rather difficult to read (small) – another font (sans serif) for the subscript letters could help.

EW and NS knife motors are clear to users of a similar microtome, but may not be self-explanatory.

Figure 1- Figure Suppl 1: Panel A: Add an arrow for the hole corresponding to the pin in Panel B.

---

## [Author Response]

Essential revisions:1) Please note the suggested expansion of figure 3 to include its own supplements 1 and 2 into the main figure for clarity of exposition and to be able to logically follow the argument. And also to relabel the axes for clarity, for example, to move the axis caption to be near the axis itself, so that the text runs nearly flush, and parallel, with the axis it captions. Otherwise, it's unclear which label corresponds to which label.

We expanded Figure 3 to include the panels from its original supplements 1 and 2. We also added an additional panel to the new Figure 3A to show the target coordinate frame alongside all the others. This should make it easier to relate the labels in panels C and D to the axes shown in A.

We also updated the axis labels for the graphs in panels E and F. The labels were moved closer, and parallel to the axes to make it easier to distinguish which label corresponds to which axis.

The only part of the original supplements 1 and 2 that wasn’t integrated into the new figure were the images of coordinate frames at different angles. These remain in the supplemental as the new, smaller Figure 3 —figure supplement 1. These were omitted from the main figure as they are not essential to understanding panels C and D, and help to reduce the number of panels in the main figure.

2) Please discuss the limitations in the ultramicrotome feed length and how these can affect automation.

We expanded the discussion to include an explanation of the feed length and its limitations (edits from line 298 – 311). We explain how the ultramicrotome must be manually reset every 200 microns (one feed length), and how this contributes to the accuracy of the workflow. We add reference to samples h/i/j in this study that all had cutting distances greater than one feed length. We also explain how further automation would require hardware improvements to increase the feed length, or allow automatic resets when the feed length is exceeded.

Reviewer #1 (Recommendations for the authors):This article is providing a critical tool to all the community.I cannot judge the geometry approach as I am not good enough in geometry to critically assess or propose any improvement in this part.Figures are of outstanding quality and make the article easy to read and interesting even for readers not directly interested in implementing the approach.I do not see at this point any need for major improvements or modifications of the article.As the limitations are explained by the authors at the end and the future developments are proposed and openly discussed, I do not have suggestions that will improve this article sufficiently to delay its publication and make a second run of revision.

Thanks to reviewer 1 for these kind comments!

Reviewer #2 (Recommendations for the authors):Enabling a solution to avoid the reset of the cutting feed (with a feed of 500 nm or less) will be a critical step towards automation. This feature will not only greatly reduce the labor requirement – freeing users from babysitting microtome during trimming, but also improve the target accuracy and operation throughput.

We agree that the need to manually reset the ultramicrotome every 200µm is a big limitation in the automation of this workflow. Unfortunately, to resolve this limitation significant hardware improvements would be necessary. Either the feed length of the ultramicrotome would have to be extended, or a method for automatic approach of knife and block face would have to be designed. As both approaches would require much development, we believe they are out of scope for this work. To reflect the importance of this issue though, we have expanded the discussion to highlight it (see comments for the second essential revision above).

Figure 3 is difficult to follow without the prelude of Figure 3—figure supplements 1 and 2. Combining these two supplement figures into Figure 3 will be more intelligible. Additionally, the axes labels in Figures 3 C and 3 D are confusing. It would be helpful to spell out in the caption which label is for x and which is for y, respectively.

See comments for the first essential revision above.

Reviewer #3 (Recommendations for the authors):The comparison between two different setups for the motorization of the microtomes is a very useful piece of data. In the description, however, the difference between the 3 motors for the Leica system and the additional translational stage motors for the RMC is not explained. It may be useful to give a simple explanation.

We expand the description in the ‘automation of the ultramicrotome’ section, to more clearly describe the differences between the motor setups in the Leica and RMC systems (edits from line 87-94). We explain how the additional translational stage motors in the RMC allow the position of the knife to be controlled in the same user interface as the orientation. We contrast this to the Leica system where translation and rotation are controlled separately through two different interfaces. In this section, we also link to the methods section ‘Ultramicrotome systems and calibration’ where the technical specifications for the motors in each system are described in detail.

The print version of the equations is rather difficult to read (small) – another font (sans serif) for the subscript letters could help.

We have increased the font size for all equations in the text to make the subscript easier to read.

EW and NS knife motors are clear to users of a similar microtome, but may not be self-explanatory.

We add the full terms ‘North-South’ and ‘East-West’ to Figure 1A and its caption, to make it clearer what these abbreviations stand for. We also add this to the first mention of NS/EW in the main text (line 88).

Figure 1- Figure Suppl 1: Panel A: Add an arrow for the hole corresponding to the pin in Panel B.

We add a label to Figure 1 —figure supplement 1 pointing to the required hole in the ultramicrotome.